# Modeling for strain-softening rocks with lateral damage based on statistical physics

**Xiaoming Li[1,2], Mingwu Wang** [ID][1]*, **Fengqiang Shen[1], Hongfei Zhang[3]**

**1** School of Civil and Hydraulic Engineering, Hefei University of Technology, Hefei, Anhui, China, **2** School of City and Architecture Engineering, Zaozhuang University, Zaozhuang, Shandong, China, **3** Shandong Zhengyuan Geological Resources Exploration Co. Ltd, Jinan, Shandong, China

* wanglab307@foxmail.com

## Abstract

Statistical physics is widely used to study the nonlinear mechanical behaviors of rock. For the limitations of existing statistical damage models and Weibull distribution, a new statistical damage with lateral damage is established. In addition, by introducing the maximum entropy distribution function and the strict constraint on damage variable, a expression of the damage variable matching the proposed model is obtained. Through comparing with the experimental results and the other two statistical damage models, the rationality of the maximum entropy statistical damage model is confirmed. The proposed model can better reflect the strain-softening behavior for rocks and respond to the residual strength, which provides a theoretical reference for practical engineering construction and design.

## 1. Introduction

Unlike most continuous materials, such as metals and plastics, rock is a geological material that contains numerous micro-pores and micro-cracks, and its mechanical progressive failure behavior is also intricate under external loads, with strong nonlinear characteristics. Since the concept of the stress-strain relationship of whole process for rocks was proposed [1], the study of rock constitutive model that reflects the stress-strain relationship of whole process has been the focus of traditional rock mechanics [2–5].

After decades of research by scholars, it is clear that the nonlinearity of the rock mechanical behaviors is closely related to the multiple damage mechanisms of its internal structure, under the external environment or loading, these damage mechanisms comprise the development and slip of primary micro-cracks, the initiation of new micro-cracks, the fragmentation and collapse of micro-pores, the elastic failure of mineral particles, the dislocation of crystals, and the clustering effect of these micro-defects. In order to reflect the constitutive relationship of rocks comprehensively, the damage mechanics began to be been introduced into the research [6, 7]. With the further development of damage mechanics for constitutive response, based on the statistical distribution characteristics of rock micro-defects, Krajcinovic [8] firstly introduced the statistics physics to simulate the stress-strain relationship of whole process for rocks, to some extent, which reveals the correlation mechanism between macroscopic phenomenology and mesoscopic damage. It should be noted that, according to

**Competing interests:** The authors have declared that no competing interests exist.

the size of the characteristic scale, the material damage can be divided into macroscopic damage, mesoscopic damage and microscopic damage. The mesoscopic damage mechanics mainly focuses on mesoscopic damage such as micro-cracks and micro-pores between macroscopic and microscopic scales, with ignoring the microscopic damage at atomic scale, such as dislocation and point defects.

The rational interpenetrating combination of statistics and damage mechanics (it is so-called statistical damage mechanics), has greatly promoted the study of progressive failure and mechanical behaviors of rocks. The constitutive models of rocks based on statistical damage theory can be roughly divided into two categories: macroscopic models and micromechanical models. The former accomplishes the damage evolution of the medium mechanical response by introducing a statistical distribution function for the failure of internal structural units into one or more sets of scalars or tensors, which is usually carried ou in the framework of continuum mechanics or irreversible thermodynamics [9–11]. The latter, namely the mesomechanical models, which are mainly composed of countless meso-elements that have a certain volume at the macro level but are small enough at the micro level, investigate the initiation and development of micro-defects at a specific scale [12–14]. In those models, the statistical distribution pattern of micro-defects corresponds to the overall damage accumulation response for rock materials.

In recent study, the failure strength of rock meso-elements is expressed in the stress or strain space based on different rock failure criteria, and it is associated with rock damage through the statistics [15, 16]. The mechanical properties of rock are directly affected by the stress level of the internal structure, and this approach is just in line with this point. However, the residual strength, that is, the strength of the softened area after the peak value of the rock, is difficult to be fully modeled, Xu [17] and Zhu [18] improved the statistical damage mode by introducing a damage correction factor, which only belonged to a mathematical concept and did not involve the mesoscopic model itself. In other similar studies, the development of rock damage is viewed as a slow process of damage accumulation, and the undamaged area would gradually change into the damaged area which still bear the load rather than become a hollow area [19–23]. Meanwhile, the statistical damage models coupled with other external environmental influences are also emerged, for example, Lin [24] suggested that the size effect should be considered in studying the failure process of rock with mesoscopic damage mechanics. In fact, the models mentioned in this paragraph still belong to the traditional elastic-plastic model with the damage variables obtained by statistics theory, more specifically, these models can also be called mesoscopic statistical damage mechanics models. The downside is that these models focus on axial damage but often ignore lateral damage in the process of model derivation, which is not very reasonable and inconsistent with the actual situation of rock deformation.

In addition, the most widely used probability density distribution function is the Weibull distribution in statistical damage models. However, Weibull distribution has its limitations, for example, the number of samples should not be less than 30 [25], the materials of rock with complex defect density or brittle [26, 27], and the probability distribution of rock strength is not approximate to the power low [28]. By contrast, the maximum entropy distribution function can overcome these problems [29, 30]. As a non-parametric probability density distribution estimation method, the maximum entropy theory can directly infer the distribution function of the parameter variables based on the information of test samples and statistical methods without assuming the distribution of the parameter variables in advance. Besides, information entropy [31] reflects the randomness of the parameter, so it is scientifically reasonable to infer the distribution function of the randomly distributed parameter variables by using the maximum entropy theory. Meanwhile, it can avoid nimiety additional personal

information. Unfortunately, the maximum entropy distribution function is rarely used to study the stress-strain behavior of rock, although Deng [32] proved the feasibility that the maximum entropy distribution function can describe rock mechanical behaviors, there is no reasonable application constraints, so there is a risk that the calculated value of damage variable may overflow, in other words, the maximum value of damage variable will exceed 1 in actual calculation, which is contrary to the physical logic.

Focusing on the above problems, this paper assumed that the strength of rock meso-elements obeys the maximum entropy distribution, and which specific probability distribution function is deduced, then the damage variable is obtained according to the statistical damage theory of rock. Furthermore, based on the statistical damage theory, a new mesoscopic statistical damage mechanics model considering the lateral damage is established, and its constitutive relation equation is deduced. Finally, the proposed model is verified by the experimental data, and compared with other statistical damage theory models. This study provides some reference significance for the study of the stress-strain whole process for rock materials.

## 2. Methodology

### 2.1 The maximum entropy principle

Information entropy is used to reflect the amount of information transferred among systems. The larger the information entropy is, the less information transmitted between systems will be; the higher the uncertainty, the greater the randomness, and vice versa. Therefore, there is a specific relationship between information entropy and the randomness of events. Let $x$ be the random variable, and $f(x)$ is the continuous probability density distribution for the random variable $x$, then its information entropy can be expressed in the form [31, 32].

$$H(x) = E[-\ln(f(x)] = -\int_R f(x)\ln(f(x))dx \tag{1}$$

where $H(x)$ is the information entropy; $E$ represents the mathematical expectation; $R$ denotes the range of $x$.

For a particular sample group, Jaynes [29, 30] holds that $f(x)$ with the maximum information entropy is the most unbiased under certain information conditions, then Eq (1) can be written as:

$$H(x)_{max} = E[-\ln(f_p(x)] = -\int_R f_p(x)\ln(f_p(x))dx \tag{2}$$

where $H(x)_{max}$ represents the maximum information entropy of $x$; $f_p(x)$ is the probability density function corresponding to $H(x)_{max}$. Just like the traditional rock statistical constitutive model, assuming that the strength of the rock meso-elements is $x$, $f_p(x)$ can be obtained by solving Eq (2) when $H(x)$ is at the maximum value. Because the maximum value of the damage variable is 1, $f_p(x)$ needs to be constrained as:

$$\int_R f_p(x) = 1 \tag{3}$$

$f_p(x)$ cannot be directly solved by Eqs (2) and (3) need to transform the direct solution problem into an optimal problem by adding constraints. Herein, the information entropy constraints mainly consist of the characteristics of the probability distribution and the statistical

characteristics of the sample data. All in all, the optimal problem can be formulated as:

$$
\begin{cases}
f_p(x) \in \arg \max H(x) = -\int_R f_p(x)\ln(f_p(x))dx \\
s.t. \int_R f_p(x) = 1 \\
\int_R g_i(x)f_p(x)dx = b_i
\end{cases}
\tag{4}
$$

where $g_i(x)$ represents the restriction function. $g_i(x)$ has an indefinite form. Deng [32] suggested its form is $x^i$ ($i = 0, 1, \ldots, m$). $b_i$ is the original moment of samples. To solve Eq (4), Eq (1) should be transformed into a Lagrange function as:

$$
H(x) = -\int_R f(x)\ln f(x)dx - \lambda_0 \int_R (f(x)dx - 1) + \sum_{i=1}^{m} \lambda_i[\int_R g_i(x)f(x)dx] - b_i
\tag{5}
$$

where $\lambda_i$ and $\lambda_0$ represent the Lagrange multipliers. If $H(x)$ is at the maximum value, the derivative of Eq (5) is required to be zero, and the result is shown as:

$$
\frac{\partial H(x)}{\partial f(x)} = 0 \Rightarrow f_p(x) = \exp(-\lambda_0 - \sum_{i=1}^{m} \lambda_i g_i(x))
\tag{6}
$$

Substituting Eq (6) into the restriction function into Eq (4), a nonlinear system of equations about the Lagrange multipliers is

$$
\int_R \exp(-\lambda_0 - \sum_{i=1}^{m} \lambda_i g_i(x))dx = 1
\tag{7}
$$

$$
\int_R g_i(x)\exp(-\lambda_0 - \sum_{i=1}^{m} \lambda_i g_i(x)) = b_i
\tag{8}
$$

The Lagrange multipliers $\lambda_0, \lambda_0, \ldots, \lambda_i$ can be derived by solving Eqs (7) and (8) with numerical solution. Next, the calculated values of $\lambda_0, \lambda_0, \ldots, \lambda_i$ are substituted into Eq (6), the specific functional form of $f_p(x)$ is determined. According to the statistical definition, the probability distribution function of the meso-elements strength for rock is written as:

$$
F(x) = \int_0^R f_p(x)dx = \int_0^R \exp(-\lambda_0 - \sum_{i=1}^{m} \lambda_i g_i(x))dx
\tag{9}
$$

where $F(\mathrm{x})$ is the probability distribution function of the meso-element strength.

## 2.2 Statistical damage evolution of rock

Suppose that the macroscopic rock is made up of $N$ meso-elements which are continuous with each other. As the stress level increases, these meso-elements are destroyed, and their number $N_d$ increases, the damage variable $D$ can be expressed as [18–20]:

$$
D = \frac{N_d}{N} = \frac{N \cdot F(x)}{N} = F(x)
\tag{10}
$$

with substituting Eq (9) into Eq (10), the damage variable with maximum entropy can be

expressed as

$$D = \int_0^R \exp\left(-\lambda_0 - \sum_{i=1}^m \lambda_i g_i(x)\right) dx \tag{11}$$

For the convenience of subsequent expression, the expression symbol of the strength for rock meso-elements is replaced by $F$, then Eq (11) is written as

$$D = \int_0^R \exp\left(-\lambda_0 - \sum_{i=1}^m \lambda_i g_i(F)\right) dF \tag{12}$$

where $D$ is the damage variable of rock with maximum entropy, $F$ is the strength of rock meso-elements.

## 3. Model establishment

### 3.1 Model derivation

The damage variable will affect the constitutive relation of rock material, which can increase the effective stress or reduce the equivalent elastic modulus. The corresponding relationship can be characterized as follows [33]

$$\sigma^* = \sigma/(1-D) = E\varepsilon/(1-D) \tag{13}$$

where $\sigma^*$ is the effective stress; $\sigma$ is the apparent stress; $E$ is the elastic modulus; $\varepsilon$ is strain; $D$ is the damage variable.

Under the external loading, an meso-element has two states of failure and non-failure [16, 17, 20, 22, 23], as shown in Fig 1. These correspond to the two states of imaginary rock

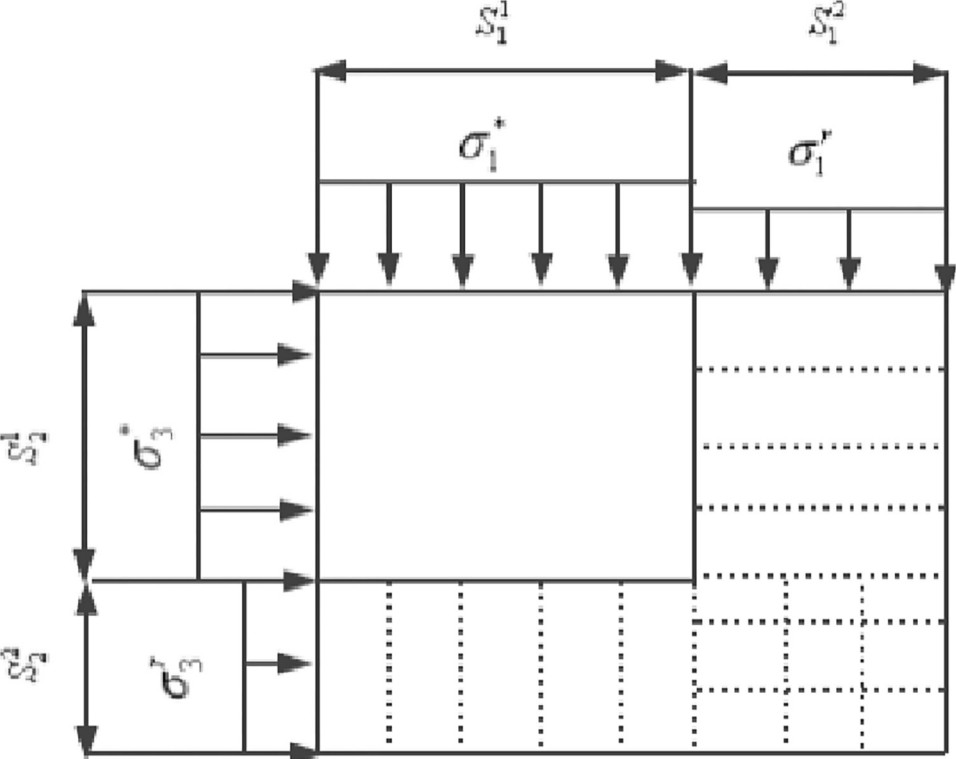

**Fig 1. The statistical damage micromechanical model.**

damaged and undamaged, respectively. $S_1^1$ and $S_1^2$ represent the non-failure and the failure area, respectively. $S_2^1$ and $S_2^2$ denotes the lateral non-failure part and the failure part, respectively. By convention, the damage variable in statistical damage mechanics is usually defined as the ratio of the failure area to the total area, so the damage variable in axial direction can be expressed in the form [20, 23].

$$D = S_1^2/(S_1^1 + S_1^2) \qquad (14)$$

where $D$ represents the damage variable in axial direction.

As shown in Fig 1, $\sigma_1^*$ and $\sigma_1'$ is the axial net stress applied on the non-failure area and the failure area, respectively; $\sigma_3^*$ and $\sigma_3'$ is the lateral net stress involved on the non-failure part and the failure part, respectively. Here, let $\sigma_1' = \sigma_1^r$, $\sigma_1^r$ is the axial residual strength, based on the static equilibrium in axial direction, we have

$$\sigma_1(S_1^1 + S_1^2) = \sigma_1^* S_1^1 + \sigma_1^r S_1^2 \qquad (15)$$

where $\sigma_1$ is the axial apparent stress. Eq (14) is substituted into Eq (15), then Eq (15) can be rewritten as

$$\sigma_1 = \sigma_1^*(1 - D) + \sigma_1^r D \qquad (16)$$

In terms of lateral damage, considering that in a triaxial test, the confining pressure remains unchanged, and when rock is completely damaged, the net stress in the damaged area should be equal to the confining pressure. Based on the above description, let $\sigma_3' = D\sigma_3$, namely, in the case of constant confining pressure, it is assumed that the net stress in the lateral damage area is proportional to the damage degree, then the lateral static equilibrium expression is written as follows

$$\sigma_3 = \sigma_3^*(1 - D) + D(\sigma_3 D) \qquad (17)$$

Eqs (16) and (17) are different from the equations derived in existing studies that ignore the lateral damage or are replaced $\sigma_3'$ with $\sigma_1^r$. By investigating Eq (17), when the damage variable gradually rises to the maximum value ($D = 1$), the lateral apparent stress value is equal to the confining pressure. It is consistent with the actual situation. Since the proposed damage mechanics model itself is a mathematical model hypothesis, there is no unified specific expression of the relationship between rock's exact mesoscopic physical damage mechanism and macroscopic phenomena. On the other hand, the primary purpose of the assumption of net stress in the lateral damage area is to solve the problem of axial residual strength. Therefore, so the expression of the mechanical mechanism of the lateral hypothesis will not be discussed in depth.

According to the mechanics of materials, each principal stress produces a linear strain in two directions besides its own for a single element body. Let $\varepsilon_1$ be the principal strain in the direction of $\sigma_1$, the expression of $\varepsilon_1$ can be obtained as

$$\varepsilon_1 = \varepsilon_1' + \varepsilon_1'' + \varepsilon_1''' \qquad (18)$$

where $\varepsilon_1'$ represents the linear strain of $\sigma_1$ in the axial direction; $\varepsilon_1''$ is the linear strain of $\sigma_2$ in the axial direction; $\varepsilon_1'''$ denotes the linear strain of $\sigma_3$ in the axial direction. Based on strain coordination, the strain produced by the undamaged material is consistent with that of the

damaged material, thus

$$\varepsilon_i^* = \varepsilon_i^r \qquad (19)$$

Combined with Eq (19) and Hooke's law, the following physical equations can be obtained as

$$\begin{cases} \varepsilon_1' = \sigma_1^*/E \\ \varepsilon_1'' = -\mu\sigma_2^*/E \\ \varepsilon_1'' = -\mu\sigma_3^*/E \end{cases} \qquad (20)$$

Substitute Eq (20) into Eq (18), the new expression of $\varepsilon_1$ is given as follows

$$\varepsilon_1 = \frac{\sigma_1^*}{E} - \frac{2\mu\sigma_3^*}{E} \qquad (21)$$

that is

$$\sigma_1^* = \varepsilon_1 E + 2\mu\sigma_3^* \qquad (22)$$

Then, substitute Eq (22) into Eq (16), the following equation is given

$$\sigma_1 = (E\varepsilon_1 + 2\mu\sigma_3^*)(1-D) + \sigma_1^r D \qquad (23)$$

Eq (17) is substituted into Eq (23), Eq (23) can be further written as:

$$\sigma_1 = E\varepsilon_1(1-D) + 2\mu\sigma_3(1-D^2) + \sigma_1^r D \qquad (24)$$

In addition, it is well-known that rock damage is closely related to the stress level, so it is necessary to determine a damage threshold to control the onset of damage. The final statistical model proposed here can be obtained by substituting Eq (12) into Eq (24), we have

$$\begin{cases} \sigma_1 = E\varepsilon_1 + 2\mu\sigma_3 + (\sigma_1^r - E\varepsilon_1)(\int_0^R \exp(-\lambda_0 - \sum_{i=1}^m \lambda_i g_i(x))dx) - 2\mu\sigma_3[\int_0^R \exp(-\lambda_0 - \sum_{i=1}^m \lambda_i g_i(x))dx]^2 \text{For F} > 0 \\ \sigma_1 = E\varepsilon_1 + 2\mu\sigma_3 \qquad\qquad \text{For} \quad F \leq 0 \end{cases} \qquad (25)$$

where $F$ represents the strength of rock meso-element. For Eq (25), when $F \leq 0$, namely no damage ($D = 0$) occurs to rocks, the constitutive model of rock will degenerate to the traditional form.

## 3.2 Determination of the strength for rock meso-elements

The strength values of the rock meso-elements are the basic for determining the probability density distribution function, which was defined in different ways. Cao [34] pointed out that the strength values ($F$) of rock meso-elements are a function of stress levels, internal friction angle, and cohesion. Pariseau [35] proposed that the failure strength of rock meso-element is closely related to rock failure criteria. In view of the good engineering practice background of the Mohr-Coulomb criteria, the suggestion of Deng [32] and Cao [31] is adopted here to depict the strength of rock meso-element, i. e.

$$F = \frac{E\varepsilon_1[(\sigma_1 - \sigma_3) - (\sigma_1 + \sigma_3)\sin\phi]}{\sigma_1 - 2\mu\sigma_3} \qquad (26)$$

where $\varphi$ is the internal friction.

### 3.3 Discussion of the axial residual strength

Existing studies on rocks mechanics behaviors have shown that the failure process of rocks is the process in which effective elements decrease and failure elements increase. At the same time, macroscopic fracture plane begins to appear slowly and gradually, leading to the gradual reduction of cohesion on the fracture plane and the gradual change of friction strength to a stable value [36], which also explains the source of rock residual strength.

As mentioned above, through the statistical damage mechanics of rock, it can be considered that the damage meso-elements can still bear a certain stress. When rock is completely damaged, the net stress of the damaged elements equals the residual strength. Here the residual strength is used to replace the net stress in the damaged area. Herein, through the proposal in Zareifard and Fahimifar [37], the axial residual strength is written as:

$$\sigma_1^r = 2c_r\cos\varphi_r/(1 - \sin\varphi_r) + \sigma_3(1 + \sin\varphi_r)/(1 - \sin\varphi_r) \tag{27}$$

where $\phi_r$ is the internal friction angle at residual strength, and $c_r$ is the cohesion at residual strength, linear regression was performed on the experimental data, then these two parameters can be obtainable.

## 4. Model validation

To verify the model proposed in this paper, the experimental data for sandstone and marble conducted by Zeng [38] and Rummel and Fairhurst [39] respectively is adopted here. Cao [34] used the experimental data of the former to verify the statistical damage model based on Weibull distribution (which is called WSDM for short here), and Li [40] used the experimental data of the latter to verify the statistical damage model based on maximum entropy distribution (which is called MSDM for short here). The proposed new statistical damage model (which is called NMSDM for short here) in this paper will be compared with these two models respectively. It should be noted that the comparison data cited from Cao [34] and Li [40] was obtained through the software: Graph Digitizer, which is a program for digitizing graphs and plots. The mechanical parameters of sandstone [38] and marble [39] are shown in Table 1.

### 4.1 K-S test of the probability distribution

The Kolmogorov-Smirnov test (K-S test) is a useful method for nonparmetric hypothesis test, which is mainly used to test whether a set of samples is derived from a probability distribution. In order to confirm the correctness of the hypothesis that the strength of rock meso-elements obey the maximum entropy distribution, so the K-S test will be carried out.

Take a set of marble samples for example, if $\sigma_2 = \sigma_3 = 3.5$MPa, through Eq (25), the strength of rock meso-elements can be calculated as (unit: MPa): 0, 9.894, 16.534, 23.920, 30.232, 38.233, 45.740, 51.433, 57.955, 64.590, 70.385, 75.594, 80.917, 80.640, 84.750, 92.551, 92.559 and 96.346, which first-order, second-order, third-order origin moments of the above samples are needed to be calculated, then according to Eq (7), the Lagrange multipliers are computed as: $\lambda_0 = 3.90680$, $\lambda_1 = 0.07510$, $\lambda_2 = -1.4\times10^{-3}$, $\lambda_3 = 5.58639\times10^{-7}$. Set the maximum entropy probability density function attained herein as $f(x)$, the corresponding probability distribution

**Table 1. Mechanical parameters of sandstone [38] and marble [39].**

| Rock classification | Elastic module (E) | Poisson's Ratio (v) | Internal friction angle (φ) |
|---|---|---|---|
| sandstone | 95 MPa | 0.2 | 31.1˚ |
| marble | 51.62 GPa | 0.25 | 44.0˚ |

**Table 2. The calculated Lagrange multipliers under different confining pressures.**

| Rock classification | Confining pressures /MPa | Lagrange multipliers | | | |
|---|---|---|---|---|---|
| | | $\lambda_0$ | $\lambda_1$ | $\lambda_2$ | $\lambda_3$ |
| Sandstone | 0 | 4.45679 | 0.02667 | $-2.053725\times10^{-4}$ | $3.95647\times10^{-7}$ |
| | 5 | 4.26732 | 0.02475 | $-2.40024\times10^{-4}$ | $6.24729\times10^{-7}$ |
| | 10 | 5.45264 | 0.02547 | $-2.84668\times10^{-5}$ | $-5.45348\times10^{-8}$ |
| | 20 | 4.57542 | 0.01621 | $-5.45348\times10^{-8}$ | $-1.98512\times10^{-8}$ |
| Marble | 3.5 | 3.90680 | 0.07510 | $-1.4\times10^{-3}$ | $5.58639\times10^{-7}$ |
| | 7 | 3.97580 | 0.04340 | $-2.0\times10^{-4}$ | $-4.15684\times10^{-7}$ |
| | 14 | 4.17670 | 0.04660 | $-5.0\times10^{-4}$ | $-1.27865\times10^{-8}$ |

function is $F(x)$, and assume $H_0$: $F(x) = F_0(x)$; Set the empirical probability distribution function as $F_n(x)$, and the statistic $D_n$ can be calculated as

$$D_n = \max_{-\infty < x < +\infty} |F_0(x) - F_n(x)| \tag{28}$$

Here, the sample size $n = 19$, and the significance level $\alpha$ is set at 0.1, the critical value $D_{19, 0.10} = 0.363$, which can be found in the K-S test table [32]. Through Eq (28), the calculated statistic $D_{19} = 0.089$, obviously $D_{19} < D_{19, 0.10}$, therefore, the original hypothesis cannot be rejected. It means that the imitative effect of the maximum entropy probability distribution function obtained is perfect.

## 4.2 Model comparison

Through the above solution method of maximum entropy function, the calculations of the Lagrange multipliers under different confining pressures are shown in Table 2.

Through substituting the Lagrange multipliers in Table 1 into Eq (25), the rock constitutive equation can be determined, and the calculation example is as follows:

The same group of samples in the above K-S test is still used for calculation demonstration, in which, the strength of meso-elements for the third point is 16.534 MPa, for facilitate understanding, the step-by-step calculation is adopted.

First, the damage variable of the third point can be given by Eq (12), we have

$$D = \int_0^{16.534} \exp(-\lambda_0 - \sum_{i=1}^{m} \lambda_i g_i(F)) dF \tag{29}$$

Second, the Lagrange multipliers ($\lambda_0 = 3.90680$, $\lambda_1 = 0.07510$, $\lambda_2 = -1.4\times10^{-3}$, $\lambda_3 = 5.58639\times10^{-7}$) calculated in this set of samples are substituted into Eq (29), then, using the software MATLAB to compile a program for integrating Eq (29), and the corresponding damage variable can be calculated to be 0.208.

Finally, by substituting the damage variable value 0.208 into Eq (24), the calculated value of stress for the third point was obtained as 69.86 MPa, and all the other points are calculated in this way. The obtained fitting curves of sandstone and marble are shown in Fig 2.

Although the physical properties and mechanical parameters of sandstone and marble differ greatly, overall, the proposed model can still simulate the post-peak softening behaviors of both, which indicates the applicability of the new model. From Fig 2, it can be found that the confining pressure influences the axial peak strength of rock and seems to increase in direct proportion, which is consistent with the traditional rock mechanics. In addition, it can be found that the greater the confining pressure, the greater the residual strength of the rock. The

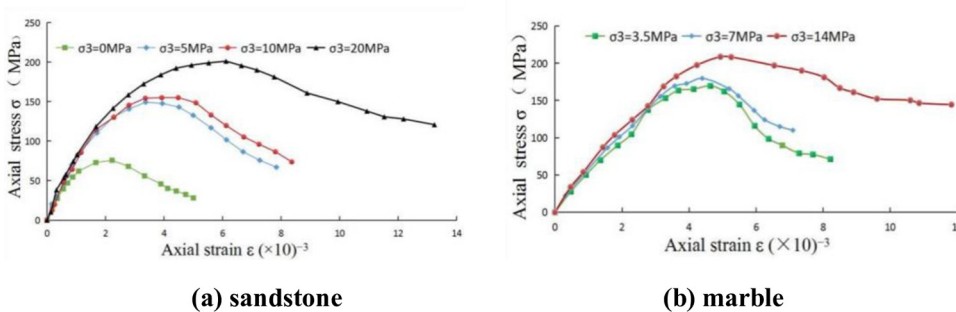

**(a) sandstone**                                                          **(b) marble**

**Fig 2.** Fitting curves of the proposed model (NMSDM): (a) sandstone; (b) marble.

probable reason is that the increase of confining pressure will increase the contact stress between each element or fracture plane inside the rock, and the residual strength will still increase when the friction coefficient remains unchanged.

The experimental curves of sandstone and marble, calculated curves of the WSDM model for sandstone and calculated curves of the MSDM for marble serve as the control groups, the comparison result is shown in Figs 3 and 4.

It is found that all these theoretical models can well reflect the progressive failure phenomenon for rocks before the peak value. In the post-peak curve stage, the axial strength gradually decreases, and the strain continues to grow, this phenomenon belongs to the strain-softening behavior of rocks, and the three calculated curves can also well capture this characteristic. At the end of the curve, the axial strength tends to a stable value, it is ostensibly independent of strain, at this point, the stress state mainly depends on the loading condition and the friction strength of the internal structure.

The actual experimental curve is not completely smooth, and there are always some abrupt points, such as the end segment of Fig 4(A), which is more obvious. However, it is difficult for the three theoretical models mentioned in this paper to deal with these irregular points accurately. Perhaps the main reason is that the statistical distribution functions used by the three theoretical models ultimately all belong to the continuous power functions in nature, the limitation of smooth statistical distribution function naturally limits the accuracy of simulation. In order to deal with these points thoroughly, it may be necessary to find more flexible statistical distribution functions or carry out piecewise simulation. Besides, it is also one of the approaches to establish a more realistic mesoscopic damage model.

## 4.3 Precision discussion

Mean relative error is a quantitative index to evaluate the simulation effect of a model, in this paper, the mean relative errors of the theoretical model for sandstone and marble are calculated respectively. It should be pointed out that the mean relative errors of the models are only used for horizontal comparison, and there is no universally accepted satisfactory value or critical value for distinguishing the simulation effect. The mean relative errors of theoretical models for sandstone and marble are given in Tables 3 and 4.

As shown in Tables 3 and 4, the proposed model can well match the experimental curves, its total mean relative errors for sandstone and marble are 10.41% and 10.12% respectively, and are slightly smaller than those of model WSDM and MSDM respectively. The mean relative error of the fitting results of the NMSDM model is smaller than the WSDM model under different confining pressures, but higher than the MSDM model when the confining pressure is 14 MPa for marble. Although the stress-strain relationship of whole process for rocks can be

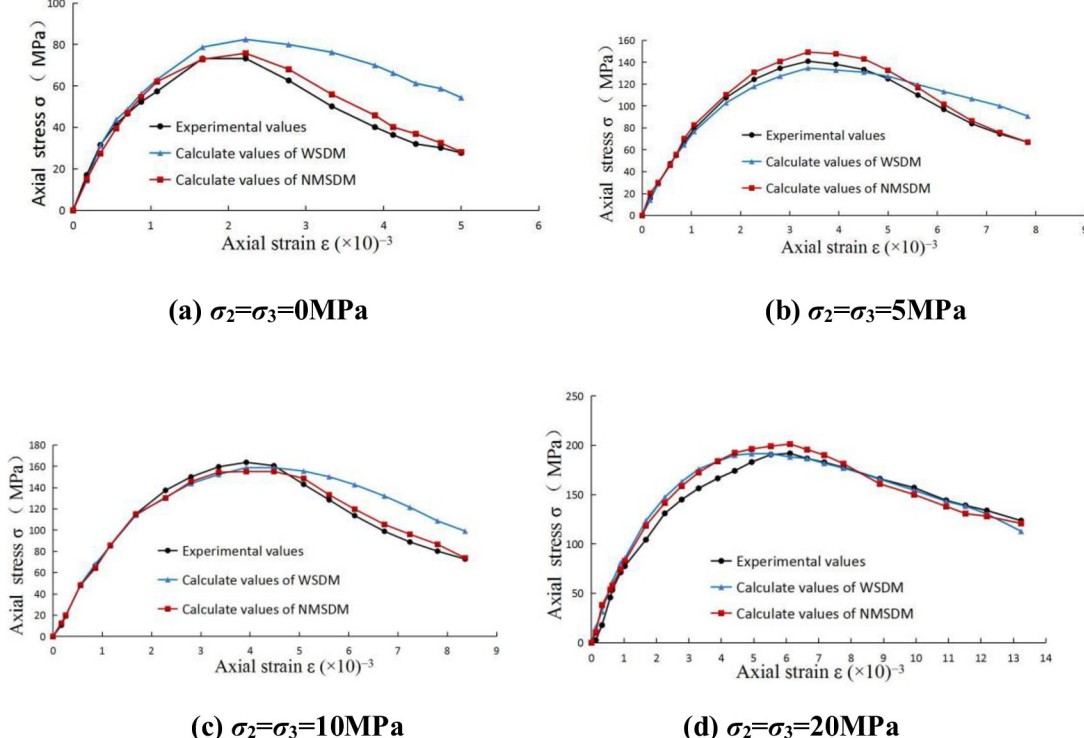

**Fig 3.** Comparison of experimental and calculated values for sandstone: (a) $\sigma_2 = \sigma_3 = 0$MPa; (b) $\sigma_2 = \sigma_3 = 5$MPa; (c) $\sigma_2 = \sigma_3 = 10$MPa; (d) $\sigma_2 = \sigma_3 = 20$MPa.

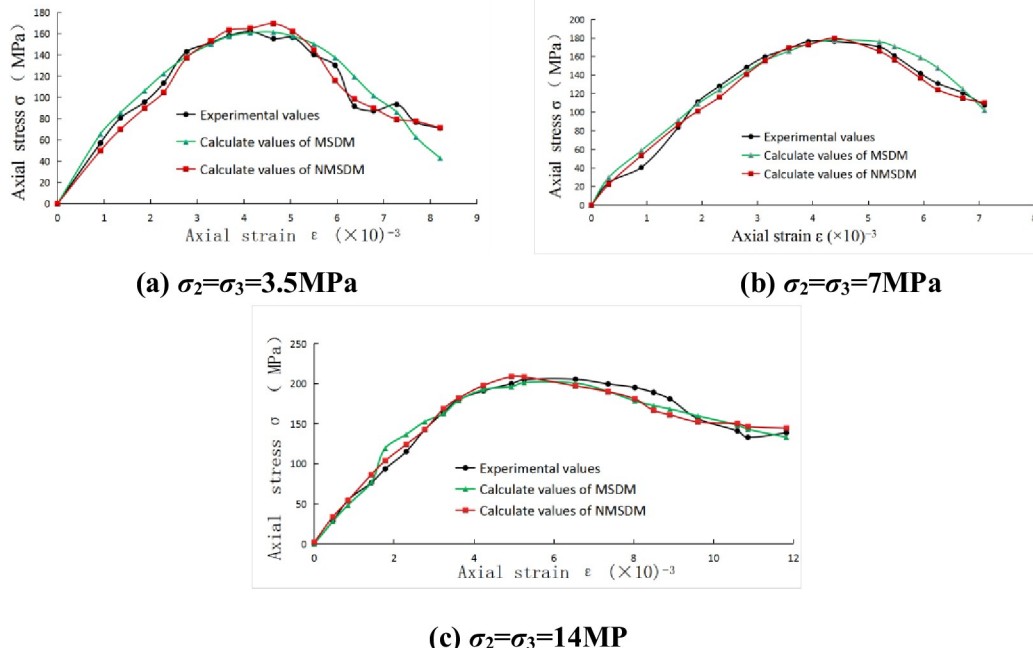

**Fig 4.** Comparison of experimental and calculated values for marble: (a) $\sigma_2 = \sigma_3 = 3.5$MPa; (b) $\sigma_2 = \sigma_3 = 7$MPa; (c) $\sigma_2 = \sigma_3 = 14$MP.

**Table 3. Mean relative errors of theoretical models for sandstone (%).**

| Model | Confining pressures /MPa | | | | Total mean relative error |
|---|---|---|---|---|---|
| | **0** | **5** | **10** | **20** | |
| WSDM | 38.80 | 9.54 | 7.38 | 31.74 | 21.87 |
| NMSDM | 7.74 | 5.64 | 4.69 | 23.60 | 10.41 |

**Table 4. Mean relative errors of theoretical models for marble (%).**

| Model | Confining pressures /MPa | | | Total mean relative error |
|---|---|---|---|---|
| | **3.5** | **7** | **14** | |
| MSDM | 15.9 | 8.43 | 6.05 | 10.12 |
| NMSDM | 8.05 | 5.89 | 6.39 | 6.77 |

reflected well by the proposed model, there are still some calculated points which deviate slightly from the experimental curve. Probably there are two reasons: From the statistical standpoint, the damage of rocks is complex, rely on a mathematical distribution function to describe the gradual failure process of rocks seem too idealistic; From the standpoint of the failure mechanism for rocks, in the gradual failure, its physical parameters will inevitably change, and it is difficult to maintain a constant, so further research is needed.

## 5. Conclusions

The exploration of rock deformation process and damage mechanism has always been the focus of geotechnical engineering research. In this research, the maximum entropy distribution function is used to describe the damage characteristics of rock, and a rock constitutive model with lateral damage is established by using meso-damage statistics theory. Using experimental data, the proposed model is compared with other theoretical models to verify its applicability. The following conclusions are obtained as follows:

1. Within a certain test range of confining pressure, the current model can respond to the mechanical behaviors of strain softening for rock. By comparison, it can be seen that the error between the theoretical results and the experimental results is relatively small, which has certain reference significance for the study of the progressive failure process of rock.

2. Compared with the other statistical damage meso-mechanics models established with the same general idea, the meso-mechanics model in this manuscripts actively considers the influence of lateral damage on the internal stress distribution of rock in conventional triaxial test, which is also an important reason why the current model can still maintain relatively good simulation accuracy at the end of the curve.

3. The maximum entropy distribution function is more objective than Weibull function which requires the experimental data to be used for the optimization and identification of parameters with a repeated inversion suspicion, but the calculations of the former are more burdensome, if for better simulation accuracy, this also can be acceptable.

4. Existing statistical damage constitutive models assume that the strain produced by the undamaged material is consistent with that of the damaged material, based on the assumptions, in the progressive failure process of rock, the physical equation of the damage material is difficult to be determined strictly, or there is a contradiction with the current statistical damage hypothesis, which requires further research and discussion.

## Supporting information

**S1 File.**
(XLS)

## Acknowledgments

Thanks are due to the three excellent reviewers and Associate Editor for their excellent and professional comments to improve the quality of the paper.

## Author Contributions

**Conceptualization:** Xiaoming Li, Mingwu Wang, Fengqiang Shen.

**Data curation:** Xiaoming Li, Hongfei Zhang.

**Formal analysis:** Mingwu Wang.

**Investigation:** Fengqiang Shen, Hongfei Zhang.

**Methodology:** Xiaoming Li.

**Resources:** Hongfei Zhang.

**Supervision:** Fengqiang Shen.

**Validation:** Xiaoming Li.

**Writing – original draft:** Xiaoming Li.

**Writing – review & editing:** Mingwu Wang.

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
