## [Decision Letter · Decision Letter 0]

27 Dec 2022

PONE-D-22-33973Modeling for strain-softening rocks with lateral damage based on statistical physicsPLOS ONE

Dear Dr. Wang,

Thank you for submitting your manuscript to PLOS ONE. After careful consideration, we feel that it has merit but does not fully meet PLOS ONE’s publication criteria as it currently stands. Therefore, we invite you to submit a revised version of the manuscript that addresses the points raised during the review process.

We look forward to receiving your revised manuscript.

Kind regards,

Jiaolong Ren

Academic Editor

PLOS ONE

Journal Requirements:

"This work was supported by the National Natural Sciences Foundation of China (No

41172274) recipient: Mingwu Wang"

"Financial support partially provided by the National Natural Sciences Foundation of China (No. 41172274) is gratefully acknowledged."

"This work was supported by the National Natural Sciences Foundation of China (No

41172274) recipient: Mingwu Wang"

Reviewers' comments:

Reviewer's Responses to Questions

**Comments to the Author**

1. Is the manuscript technically sound, and do the data support the conclusions?

Reviewer #1: Yes

Reviewer #2: Yes

Reviewer #3: No

2. Has the statistical analysis been performed appropriately and rigorously? 

Reviewer #1: Yes

Reviewer #2: Yes

Reviewer #3: No

3. Have the authors made all data underlying the findings in their manuscript fully available?

Reviewer #1: Yes

Reviewer #2: Yes

Reviewer #3: Yes

4. Is the manuscript presented in an intelligible fashion and written in standard English?

Reviewer #1: Yes

Reviewer #2: Yes

Reviewer #3: No

5. Review Comments to the Author

Reviewer #1: A new statistical damage that combining the maximum entropy distribution function and the strict constraint on damage variable with lateral damage is established. The viewpoint of this paper is novelty enough, however, the more examples need to be added to verify. In addition, why is Eq (9) choosen as the distribution function?

Reviewer #2: In this manuscript, a new statistical constitutive model considering the lateral damage is proposed based on the meso-damage statistics theory, and its constitutive relation equation is deduced. Furthermore, using experimental data, the proposed model is verified by comparing the Weibull model.

The paper is well organized, however, there are some modifications before publication in PLOS ONE.

1.It is suggested that Eq. (14) should be rewritten to avoid misunderstanding.

2. Please check whether Eq. (17) is correct?

3. The names of Fig 3b,3c and 3d are wrong, please modify it.

4. It is suggested that more examples are used to validate the validity of the proposed model.

Reviewer #3: The non-linear mechanical characteristics of rocks were studied on the basis of statistical physics. A model was proposed, which as stated in the manuscript, can predict the strain-softening behavior for rocks better and can respond to the residual strength.

The presentation style of the manuscript was not good and should be improved. There is a lack of literature in the introduction section and the aim of the study was not clearly identified, showing the gap in the literature. The discussion of the manuscript is very general and common. The main part of the manuscript is the model which is not clear what are the concepts beyond the assumptions made, i.e. how damage/failure is linked to the elasticity theory. Thus, I would consider the contribution of this work to be marginal and this work is not valid for publication.

Below are several comments that can be useful for the authors to improve the manuscript.

• Difficult to identify the exact position of the comments due to the missing of the line numbering.

• Please refer to the underlined parts in the main manuscript “the pdf version” when reading the comments below.

• Introduction:

1. Overall, there is a lack in the previous studies and literature in the “Introduction” section. Therefore, it is highly recommended to extend the section in order to study the gaps in the literature and the aims of the paper can be clearly explained.

2. Please explain further how/and why rocks are more complex materials are compared with other materials.

3. Please explain how the complexity of the failure mechanism related strongly to the non-linear stress-strain relationship.

4. Please explain further the term “mesoscopic damage” should be defined in the text.

5. The authors used the term “residual strength”, however, the term should be defined clearly and should be distinguished from the “Critical state”.

6. In page 3, please explain what are the problems associated with the modified models, so that they should be clear for the readers.

7. What is the parameter” D” in Page 3.

• Section “Establishment of the Statistical damage model”

8. While the authors tried to derive their model which presents “damage”, it was not clear how the damage or failure of a rock sample was related to the modulus of elasticity and Passions ratio. In other words, how the failure model was related to the zone of elasticity, where the sample can recover.

It may be unrealistic to rely upon the elasticity parameters to reflect/interpret the damage behavior of rocks.

• Section “Model Validation”

9. Page 11: What do the authors by the K-S model? The K-S Model should be first explained as it was introduced to the reader for the first time.

10. Not The conclusions made in Page 13 are very general and common in the literature.

11. Not clear how the data of Weibull model in Table 2 were obtained. Need to be explained/clarified.

12. The conclusions made in Page 14 are also very general and common in the literature.

13. The proposed model validated against the experimental data with a maximum mean relative error of 10.41%. To what extent, this error satisfactory? What is the range of acceptance? Give reference(s) if possible.

14. It was not clear how the models produced the stress-strain curves in Figs. 2 and 3. This should be explained; a clarification example may be helpful.

15. The proposed mode was only validated against one set of data, apart from the ambiguity, explained in point 14 above. The model, therefore, needs to be validated against many experimental results.

16. It is highly recommended to reword the conclusion section.

17. Please provide figure captions for the figures.

6. PLOS authors have the option to publish the peer review history of their article (what does this mean?). If published, this will include your full peer review and any attached files.

Reviewer #1: No

Reviewer #2: **Yes: **Qiang Zhang

Reviewer #3: No

---

## [Author Response · Author response to Decision Letter 0]

19 Feb 2023

Reviewers' comments:

Reviewer's Responses to Questions

Comments to the Author

1. Is the manuscript technically sound, and do the data support the conclusions?

Reviewer #1: Yes

Reviewer #2: Yes

Reviewer #3: No

Thank you very much for suggestions. This paper assumed that the strength of rock meso-elements obeys the maximum entropy distribution, and which specific probability distribution function is deduced, then the damage variable is obtained according to the statistical damage theory of rock. Furthermore, based on the statistical damage theory, a new mesoscopic statistical damage mechanics model considering the lateral damage is established, and its constitutive relation equation is deduced. Finally, the proposed model is verified by the experimental data, and compared with other statistical damage theory models. The data have been provided as supporting information.

2. Has the statistical analysis been performed appropriately and rigorously?

Reviewer #1: Yes

Reviewer #2: Yes

Reviewer #3: No

Thank you very much for your suggestions. In order to verify the validity of the theoretical model, the theoretical simulation was carried out on two different rocks, sandstone and marble respectively, and multiple sets of data are simulated for each rock. In addition, it also makes full comparison with other theoretical models. The accuracy of the model is verified by the basic mechanical behaviors of rock, and a reasonable explanation is given for the unconventional simulation points. In the process of data statistical analysis, objective and unbiased calculation methods are adopted as far as possible to avoid subjective interference. 

3. Have the authors made all data underlying the findings in their manuscript fully available?

Reviewer #1: Yes

Reviewer #2: Yes

Reviewer #3: Yes

Thank you very much for suggestions. The data have been provided as supporting information.

4. Is the manuscript presented in an intelligible fashion and written in standard English?

Reviewer #1: Yes

Reviewer #2: Yes

Reviewer #3: No

Thank you very much for suggestions. Our manuscript is further polished by the coauthor Dr. Shen who studied and worked abroad for more than ten years in the United Kingdom. 

5.Review Comments to the Author

Reviewer #1: A new statistical damage that combining the maximum entropy distribution function and the strict constraint on damage variable with lateral damage is established. The viewpoint of this paper is novelty enough, however, the more examples need to be added to verify. In addition, why is Eq.(9) choosen as the distribution function?

Due to the submission system, some formulas and pictures need to be viewed in the document "Response to Reviewers".

Thank you very much for your suggestions. 

We have added three sets of experimental data for marble to verify the proposed model in the revised manuscript, and the added content is mainly in the section“Model Validation”.

why is Eq.(9) choosen as the distribution function?

Thank you very much for your suggestions. 

Eq.(9) is a statistical distribution function deduced by using the maximum entropy principle based on assuming that the probability distribution of the meso-elements strength has statistical characteristics. The maximization of entropy under moment constraints leads to a least biased estimate of probability density function, which covers most empirical probability distributions as special cases. Compared with the classic random distribution, the maximum entropy distribution function can make full use of the sample information without excessively depending on the sample, which has more sufficient mathematical and physical significance. Which is also an important reason why we choose Eq.(9) as the statistical distribution function.

Reviewer #2: In this manuscript, a new statistical constitutive model considering the lateral damage is proposed based on the meso-damage statistics theory, and its constitutive relation equation is deduced. Furthermore, using experimental data, the proposed model is verified by comparing the Weibull model.The paper is well organized, however, there are some modifications before publication in PLOS ONE.

Due to the submission system, some formulas and pictures need to be viewed in the document "Response to Reviewers".

1.It is suggested that Eq. (14) should be rewritten to avoid misunderstanding.

Thank you very much for your suggestions. We have rewritten Eq. (14), and the origin of Eq. (14) is further explained, the modified Eq. (14) is as follows:

 (14)

where D represents the damage variable in axial direction. It should be pointed out that the number of failure meso-elements of rock is directly related to the area of the damaged part, so Eq. (12) and Eq. (14) are equal [18-20,23].

2.Please check whether Eq. (17) is correct?

Thank you very much for your suggestions, we have checked it, in order to avoid misunderstanding, the modified Eq. (17) is rewritten as follows:

 (17)

3.The names of Fig 3b,3c and 3d are wrong, please modify it.

Thank you very much for your suggestions, we have modified it as follows:

Figure 3. Comparison of experimental and calculated values for sanstone: (a) σ2=σ3=0MPa; (b) σ2=σ3=5MPa; (c) σ2=σ3=10MPa; (d) σ2=σ3=20MPa.

4.It is suggested that more examples are used to validate the validity of the proposed model.

Thank you very much for your suggestions. We have added more examples to validate the validity of the proposed model. Three sets of experimental data for marble is added to verify the suitability of the current model. The added content is mainly in the section“Model Validation”, part of the calculated results of the added examples are shown as follows:

Reply to Reviewer 3

Reviewer #3: The non-linear mechanical characteristics of rocks were studied on the basis of statistical physics. A model was proposed, which as stated in the manuscript, can predict the strain-softening behavior for rocks better and can respond to the residual strength.

The presentation style of the manuscript was not good and should be improved. There is a lack of literature in the introduction section and the aim of the study was not clearly identified, showing the gap in the literature. The discussion of the manuscript is very general and common. The main part of the manuscript is the model which is not clear what are the concepts beyond the assumptions made, i.e. how damage/failure is linked to the elasticity theory. Thus, I would consider the contribution of this work to be marginal and this work is not valid for publication.

Below are several comments that can be useful for the authors to improve the manuscript.

• Difficult to identify the exact position of the comments due to the missing of the line numbering.

• Please refer to the underlined parts in the main manuscript “the pdf version” when reading the comments below.

Due to the submission system, some formulas and pictures need to be viewed in the document "Response to Reviewers".

Thank you for your suggestions, and thank you for spending your precious time to review our manuscript sincerely. The careful review marks in the revised draft fully demonstrate your rigorous working attitude, we are also sorry for the low-level mistakes in original manuscript. Your suggestions are of great significance to this paper. We will revised the manuscript carefully and seriously, the responses to reviews

are listed point by point as follows:

Introduction:

1. Overall, there is a lack in the previous studies and literature in the “Introduction” section. Therefore, it is highly recommended to extend the section in order to study the gaps in the literature and the aims of the paper can be clearly explained.

Thank you very much for your suggestions. We have integrated and rewritten the "Introduction", and the modified part is marked in colored red fonts. The modified content is as follows:

Introduction

Unlike most continuous materials, such as metals and plastics, rock is a geological material that contains numerous micro-pores and micro-cracks, and its mechanical progressive failure behavior is also intricate under external loads, with strong nonlinear characteristics. Since the concept of the stress-strain relationship of whole process for rocks was proposed [1], the study of rock constitutive model that reflects the stress-strain relationship of whole process has been the focus of traditional rock mechanics [2–5].

After decades of research by scholars, it is clear that the nonlinearity of the rock mechanical behaviors is closely related to the multiple damage mechanisms of its internal structure, under the external environment or loading, these damage mechanisms comprise the development and slip of primary micro-cracks, the initiation of new micro-cracks, the fragmentation and collapse of micro-pores, the elastic failure of mineral particles, the dislocation of crystals, and the clustering effect of these micro-defects. In order to reflect the constitutive relationship of rocks comprehensively, the damage mechanics began to be been introduced into the research[6-7]. With the further development of damage mechanics for constitutive response, based on the statistical distribution characteristics of rock micro-defects, Krajcinovic [8] firstly introduced the statistics physics to simulate the stress-strain relationship of whole process for rocks, to some extent, which reveals the correlation mechanism between macroscopic phenomenology and mesoscopic damage. It should be noted that, according to the size of the characteristic scale, the material damage can be divided into macroscopic damage, mesoscopic damage and microscopic damage. The mesoscopic damage mechanics mainly focuses on mesoscopic damage such as micro-cracks and micro-pores between macroscopic and microscopic scales, with ignoring the microscopic damage at atomic scale, such as dislocation and point defects. 

The rational interpenetrating combination of statistics and damage mechanics (it is so-called statistical damage mechanics), has greatly promoted the study of progressive failure and mechanical behaviors of rocks. The constitutive models of rocks based on statistical damage theory can be roughly divided into two categories: macroscopic models and micromechanical models. The former accomplishes the damage evolution of the medium mechanical response by introducing a statistical distribution function for the failure of internal structural units into one or more sets of scalars or tensors, which is usually carried ou in the framework of continuum mechanics or irreversible thermodynamics[9-11]. The latter, namely the mesomechanical models, which are mainly composed of countless meso-elements that have a certain volume at the macro level but are small enough at the micro level, investigate the initiation and development of micro-defects at a specific scale[12-14]. In those models, the statistical distribution pattern of micro-defects corresponds to the overall damage accumulation response for rock materials.

In recent study, the failure strength of rock meso-elements is expressed in the stress or strain space based on different rock failure criteria, and it is associated with rock damage through the statistics[15,16]. The mechanical properties of rock are directly affected by the stress level of the internal structure, and this approach is just in line with this point. However, the residual strength, that is, the strength of the softened area after the peak value of the rock, is difficult to be fully modeled, Xu [17] and Zhu [18] improved the statistical damage mode by introducing a damage correction factor, which only belonged to a mathematical concept and did not involve the mesoscopic model itself. In other similar studies, the development of rock damage is viewed as a slow process of damage accumulation, and the undamaged area would gradually change into the damaged area which still bear the load rather than become a hollow area[19-23]. Meanwhile, the statistical damage models coupled with other external environmental influences are also emerged, for example, Lin [24] suggested that the size effect should be considered in studying the failure process of rock with mesoscopic damage mechanics. In fact, the models mentioned in this paragraph still belong to the traditional elastic-plastic model with the damage variables obtained by statistics theory, more specifically, these models can also be called mesoscopic statistical damage mechanics models. The downside is that these models focus on axial damage but often ignore lateral damage in the process of model derivation, which is not very reasonable and inconsistent with the actual situation of rock deformation.

In addition, the most widely used probability density distribution function is the Weibull distribution in statistical damage models. However, Weibull distribution has its limitations , for example, the number of samples should not be less than 30 [25], the materials of rock with complex defect density or brittle [26,27], and the probability distribution of rock strength is not approximate to the power low [28]. By contrast, the maximum entropy distribution function can overcome these problems [29,30]. As a non-parametric probability density distribution estimation method, the maximum entropy theory can directly infer the distribution function of the parameter variables based on the information of test samples and statistical methods without assuming the distribution of the parameter variables in advance. Besides, information entropy [31] reflects the randomness of the parameter, so it is scientifically reasonable to infer the distribution function of the randomly distributed parameter variables by using the maximum entropy theory. Meanwhile, it can avoid nimiety additional personal information. Unfortunately, the maximum entropy distribution function is rarely used to study the stress-strain behavior of rock, although Deng [32] proved the feasibility that the maximum entropy distribution function can describe rock mechanical behaviors, there is no reasonable application constraints, so there is a risk that the calculated value of damage variable may overflow, in other words, the maximum value of damage variable will exceed 1 in actual calculation, which is contrary to the physical logic.

Focusing on the above problems, this paper assumed that the strength of rock meso-element obeys the maximum entropy distribution, and which specific probability distribution function is deduced, then the damage variable is obtained according to the statistical damage theory of rock. Furthermore, based on the statistical damage theory, a new mesoscopic statistical damage mechanics model considering the lateral damage is established, and its constitutive relation equation is deduced. Finally, the proposed model is verified by the experimental data, and compared with other statistical damage theory models. This study provides some reference significance for the study of the stress-strain whole process for rock materials.

2. Please explain further how/and why rocks are more complex materials are compared with other materials.

Thank you very much for your suggestions. We have further explained it in Lines 25-26 of section “Introduction”, the specific modified content is as follows:

“Unlike most continuous materials, such as metals and plastics, rock is a geological material that contains numerous micro-pores and micro-cracks”.

3. Please explain how the complexity of the failure mechanism related strongly to the non-linear stress-strain relationship.

Thank you very much for your suggestions. The damage mechanism is multifaceted , which comprehensively affects the mechanical behaviors of rock. We have further elaborated on this in Lines 31-36 as follow:

“After decades of research by scholars, it is clear that the nonlinearity of the rock mechanical behaviors is closely related to the multiple damage mechanisms of its internal structure, under the external environment or loading, these damage mechanisms comprise the development and slip of primary micro-cracks, the initiation of new micro-cracks, the fragmentation and collapse of micro-pores, the elastic failure of mineral particles, the dislocation of crystals, and the clustering effect of these micro-defects”.

4. Please explain further the term “mesoscopic damage” should be defined in the text.

Thank you very much for your suggestions. We have explained further the term “mesoscopic damage”in Lines 42-47of revised version. The added elaboration is as follows:

“It should be noted that, according to the size of the characteristic scale, the material damage can be divided into macroscopic damage, mesoscopic damage and microscopic damage. The mesoscopic damage mechanics mainly focuses on mesoscopic damage such as micro-cracks and micro-pores between macroscopic and microscopic scales, with ignoring the microscopic damage at atomic scale, such as dislocation and point defects”.

5. The authors used the term “residual strength”, however, the term should be defined clearly and should be distinguished from the “Critical state”.

Thank you very much for your suggestions. This term has been further defined in Lines 64-65 , which is as follows:

“ However, the residual strength, that is, the strength of the softened area after the peak value of the rock”

6. In page 3, please explain what are the problems associated with the modified models, so that they should be clear for the readers.

Thank you very much for your suggestions. This paragraph has been rewritten as a whole, and the corresponding modified content in Lines 76-79 is as follows:

“ The downside is that these models focus on axial damage but often ignore lateral damage in the process of model derivation, which is not very reasonable and inconsistent with the actual situation of rock deformation”.

7. What is the parameter” D” in Page 3.

Thank you very much for your suggestions. “D” represents the damage variable in the original text, but in the revised version, this academic symbol has been removed from the introduction. Here the content is reorganized in Lines 95-97 as follows:.

“so there is a risk that the calculated value of damage variable may overflow, in other words, the maximum value of damage variable will exceed 1 in actual calculation, which is contrary to the physical logic.”

Section “Establishment of the Statistical damage model”

8. While the authors tried to derive their model which presents “damage”, it was not clear how the damage or failure of a rock sample was related to the modulus of elasticity and Passions ratio. In other words, how the failure model was related to the zone of elasticity, where the sample can recover.

It may be unrealistic to rely upon the elasticity parameters to reflect/interpret the damage behavior of rocks.

Thank you very much for your suggestions. Before explaining the above problems, please allow us briefly to go over the derivation of the proposed model in our paper again:

Fig 1. The statistical damage micromechanical model

First of all:

As can be seen in Fig 1, suppose that rocks consist of many mesoscopic elements, whose properties vary from one element to another. A mesoscopic element has dual states: available or failed. The two types of elements represent the fictitious undamaged and damaged configurations, respectively. 

Based on the above assumptions, we define: and represent the non-failure and the failure area, respectively. and denotes the lateral non-failure part and the failure part, respectively. and is the axial net stress applied on the non-failure area and the failure area, respectively; and is the lateral net stress involved on the non-failure part and the failure part, respectively. Here, let , is the axial residual strength, With the increase of axial load, the damage area will gradually become larger, so the damage variable is defined as:

 (14)

The definition of damage variable here is similar to the classical definition of damage variable by effective stress area, however, the former thinks that the meso-elements in the damaged area can still bear a certain stress, while the latter does not pay attention to this.

Based on the static equilibrium in axial direction, we have

 (15)

By default, during the process of axial load increase, the bearing area of rock remains unchanged, but the area of damaged part and undamaged part are constantly transformed. 

Eq. (14) is substituted into Eq. (15), then Eq. (15) can be rewritten as

 (16)

Secondly:

According to the generalized Hooke's law in mechanics of materials, the familiar equation can be obtained as:

 (22)

Then, substitute Eq. (22) into Eq. (16), the following equation is given

 (23)

At this point, the damage variable is introduced into the elastic-plastic equation. Of course, in fact, this is only a mathematical combination, not enough to explain how the damage or failure of a rock sample was related to the modulus of elasticity and Passions ratio.

 Let's look at Eq. (16) again, on the face of it, the damage variable is not directly related to the the modulus of elasticity and Passions ratio. In the section “Methodology”, the trend function of damage variable is derived by using the strength of meso-elements as the calculation samples, and the the strength of meso-elements needs to be calculated according to the failure criterion of rock.

 (26)

Discussion:

From Eq. (16), there is a certain correlation between the damage variable and rock mechanical parameters. If inspect Eq. (23) alone, it seems that the damage variable does play a role in reducing mechanical parameters, which is a kind of coincidence. After all, the damage hypothesis in this paper is mainly based on the effective stress principle, however, there are some literatures who regard the result of elastic modulus change as equivalent elastic modulus. Our reply is not comprehensive, and we will continue to conduct more in-depth research on this issue in the future.

Section “Model Validation”

9. Page 11: What do the authors by the K-S model? The K-S Model should be first explained as it was introduced to the reader for the first time.

Thank you very much for your suggestions. We have modified it in Lines 277-279:

“The Kolmogorov-Smirnov test ( K-S test ) is a useful method for nonparmetric hypothesis test, which is mainly used to test whether a set of samples is derived from a probability distribution. In order to confirm the correctness of the hypothesis that the strength of rock meso-elements obey the maximum entropy distribution, so the K-S test will be carried out.”

10. Not The conclusions made in Page 13 are very general and common in the literature.

Thank you very much for your suggestions. We have rewritten the conclusions to certain extent, which are in Lines 316-320.

“Although the physical properties and mechanical parameters of sandstone and marble differ greatly, overall, the proposed model can still simulate the post-peak softening behaviors of both, which indicates the applicability of the new model. From Fig 2, it can be found that the confining pressure influences the axial peak strength of rock and seems to increase in direct proportion, which is consistent with the traditional rock mechanics.”

11. Not clear how the data of Weibull model in Table 2 were obtained. Need to be explained/clarified.

Thank you very much for your suggestions, we have explained that in Lines 270-272:

“It should be noted that the comparison data cited from Cao [34] and Li [40] was obtained through the software: Graph Digitizer, which is a program for digitizing graphs and plots.”

12. The conclusions made in Page 14 are also very general and common in the literature.

Thank you very much for your suggestions, We have rewritten the conclusions to certain extent, and added some new conclusions in Lines 333-348.

“It is found that all these theoretical models can well reflect the progressive failure phenomenon for rocks before the peak value. In the post-peak curve stage, the axial strength gradually decreases, and the strain continues to grow, this phenomenon belongs to the strain-softening behavior of rocks, and the three calculated curves can also well capture this characteristic. At the end of the curve, the axial strength tends to a stable value, it is ostensibly independent of strain, at this point, the stress state mainly depends on the loading condition and the friction strength of the internal structure.

The actual experimental curve is not completely smooth, and there are always some abrupt points, such as the end segment of Fig 4 (a), which is more obvious. However, it is difficult for the three theoretical models mentioned in this paper to deal with these irregular points accurately. Perhaps the main reason is that the statistical distribution functions used by the three theoretical models ultimately all belong to the continuous power functions in nature, the limitation of smooth statistical distribution function naturally limits the accuracy of simulation. In order to deal with these points thoroughly, it may be necessary to find more flexible statistical distribution functions or carry out piecewise simulation. Besides, it is also one of the approaches to establish a more realistic mesoscopic damage model.”

13. The proposed model validated against the experimental data with a maximum mean relative error of 10.41%. To what extent, this error satisfactory? What is the range of acceptance? Give reference(s) if possible.

Thank you very much for your suggestions, we have explained that in Lines 352-354:

“It should be pointed out that the mean relative errors of the models are only used for horizontal comparison, and there is no universally accepted satisfactory value or critical value for distinguishing the simulation effect.”

14. It was not clear how the models produced the stress-strain curves in Figs. 2 and 3. This should be explained; a clarification example may be helpful.

Thank you very much for your suggestions, a clarification example in the revised manuscript is given as:

The same group of samples in the above K-S test is still used for calculation demonstration, in which, the strength of meso-elements for the third point is 16.534 MPa, for facilitate understanding, the step-by-step calculation is adopted. 

First, the damage variable of the third point can be given by Eq. (12), we have 

 (29)

Second, the Lagrange multipliers (λ0=3.90680, λ1=0.07510, λ2=-1.4×10-3, λ3=5.58639×10-7) calculated in this set of samples are substituted into Eq. (29), then, using the software MATLAB to compile a program for integrating Eq. (29), and the corresponding damage variable can be calculated to be 0.208.

Finally, by substituting the damage variable value 0.208 into Eq. (24), the calculated value of stress for the third point was obtained as 69.86 MPa, and all the other points are calculated in this way. The obtained fitting curves of sandstone and marble are shown in Fig 2. 

15. The proposed mode was only validated against one set of data, apart from the ambiguity, explained in point 14 above. The model, therefore, needs to be validated against many experimental results.

Thank you very much for your suggestions. We have added more examples to validate the validity of the proposed model. Three sets of experimental data for marble is added to verify the suitability of the current model. The added content is mainly in the section“Model Validation”, part of the calculated results of the added examples are shown as follows:

(a) sandstone (b) marble

Fig 2. Fitting curves of the proposed model (NMSDM): (a) sandstone; (b) marble.

 (a) σ2=σ3=3.5MPa (b) σ2=σ3=7MPa

(c) σ2=σ3=14MP

Figure 4. Comparison of experimental and calculated values for marble: (a) σ2=σ3=3.5MPa; (b) σ2=σ3=7MPa; (c) σ2=σ3=14MP.

16. It is highly recommended to reword the conclusion section.

Thank you very much for your suggestions. We have reword the conclusion section, which is shown as follows:

Conclusions

The exploration of rock deformation process and damage mechanism has always been the focus of geotechnical engineering research. In this research, the maximum entropy distribution function is used to describe the damage characteristics of rock, and a rock constitutive model with lateral damage is established by using meso-damage statistics theory. Using experimental data, the proposed model is compared with other theoretical models to verify its applicability. The following conclusions are obtained as follows:

1)Within a certain test range of confining pressure, the current model can respond to the mechanical behaviors of strain softening for rock. By comparison, it can be seen that the error between the theoretical results and the experimental results is relatively small, which has certain reference significance for the study of the progressive failure process of rock.

2)Compared with the other statistical damage meso-mechanics models established with the same general idea, the meso-mechanics model in this manuscripts actively considers the influence of lateral damage on the internal stress distribution of rock in conventional triaxial test, which is also an important reason why the current model can still maintain relatively good simulation accuracy at the end of the curve. 

3)The maximum entropy distribution function is more objective than Weibull function which requires the experimental data to be used for the optimization and identification of parameters with a repeated inversion suspicion, but the calculations of the former are much more burdensome, if for better simulation accuracy, this also can be acceptable.

17. Please provide figure captions for the figures.

Thank you very much for your suggestions. We have provide figure captions for the figures in the revised manuscript

---

## [Decision Letter · Decision Letter 1]

6 Mar 2023

Modeling for strain-softening rocks with lateral damage based on statistical physics

PONE-D-22-33973R1

Dear Dr. Mingwu Wang,

We're pleased to inform you that your manuscript has been judged scientifically suitable for publication and will be formally accepted for publication once it meets all outstanding technical requirements.

Within one week, you'll receive an e-mail detailing the required amendments. When these have been addressed, you'll receive a formal acceptance letter and your manuscript will be scheduled for publication.

Kind regards,

Jiaolong Ren

Academic Editor

PLOS ONE

Additional Editor Comments (optional):

Reviewers' comments:

Reviewer's Responses to Questions

**Comments to the Author**

1. If the authors have adequately addressed your comments raised in a previous round of review and you feel that this manuscript is now acceptable for publication, you may indicate that here to bypass the “Comments to the Author” section, enter your conflict of interest statement in the “Confidential to Editor” section, and submit your "Accept" recommendation.

Reviewer #1: (No Response)

Reviewer #2: All comments have been addressed

Reviewer #3: All comments have been addressed

2. Is the manuscript technically sound, and do the data support the conclusions?

Reviewer #1: (No Response)

Reviewer #2: Yes

Reviewer #3: Yes

3. Has the statistical analysis been performed appropriately and rigorously? 

Reviewer #1: (No Response)

Reviewer #2: Yes

Reviewer #3: Yes

4. Have the authors made all data underlying the findings in their manuscript fully available?

Reviewer #1: (No Response)

Reviewer #2: Yes

Reviewer #3: Yes

5. Is the manuscript presented in an intelligible fashion and written in standard English?

Reviewer #1: (No Response)

Reviewer #2: Yes

Reviewer #3: Yes

6. Review Comments to the Author

Reviewer #1: (No Response)

Reviewer #2: All my comment have been addressed properly, and I have no more comments.

Reviewer #3: (No Response)

7. PLOS authors have the option to publish the peer review history of their article (what does this mean?). If published, this will include your full peer review and any attached files.

Reviewer #1: No

Reviewer #2: **Yes: **Qiang Zhang

Reviewer #3: No

---

## [Editor Report · Acceptance letter]

21 Mar 2023

PONE-D-22-33973R1 

Modeling for strain-softening rocks with lateral damage based on statistical physics 

Dear Dr. Wang:

I'm pleased to inform you that your manuscript has been deemed suitable for publication in PLOS ONE. Congratulations! Your manuscript is now with our production department. 

Kind regards, 

on behalf of

Dr. Jiaolong Ren 

Academic Editor

PLOS ONE